# Unveiling Antibacterial Potential and Physiological Characteristics of Thermophilic Bacteria Isolated from a Hot Spring in Iran

**DOI:** 10.3390/microorganisms12040834

**Published:** 2024-04-20

**Authors:** Zeinab Rafiee, Maryam Jalili Tabaii, Maryam Moradi, Sharareh Harirchi

**Affiliations:** 1Department of Biotechnology, Faculty of Biological Sciences and Technology, Shahid Ashrafi Esfahani University, Isfahan 81799-49999, Iran; zeinabrf731@gmail.com (Z.R.); maryam.moradi5182@gmail.com (M.M.); 2Swedish Centre for Resource Recovery, University of Borås, 501 90 Borås, Sweden; 3Department of Biotechnology, Iranian Research Organization for Science and Technology, Tehran P.O. Box 3353-5111, Iran

**Keywords:** antimicrobial resistance, pathogens, *Bacillus*, actinomycetes, bioactive materials, extreme conditions

## Abstract

The increasing worldwide demand for antimicrobial agents has significantly contributed to the alarming rise of antimicrobial resistance, posing a grave threat to human life. Consequently, there is a pressing need to explore uncharted environments, seeking out novel antimicrobial compounds that display exceptionally efficient capabilities. Hot springs harbor microorganisms possessing remarkable properties, rendering them an invaluable resource for uncovering groundbreaking antimicrobial compounds. In this study, thermophilic bacteria were isolated from Mahallat Hot Spring, Iran. Out of the 30 isolates examined, 3 strains exhibited the most significant antibacterial activities against *Escherichia coli* and *Staphylococcus aureus*. Furthermore, the supernatants of the isolated strains exhibited remarkable antibacterial activity, displaying notable resistance to temperatures as high as 75 °C for 30 min. It was determined that the two strains showed high similarity to the *Bacillus* genus, while strain Kh3 was classified as *Saccharomonospora azurea*. All three strains exhibited tolerance to NaCl. *Bacillus* strains demonstrated optimal growth at pH 5 and 40 °C, whereas *S. azurea* exhibited optimal growth at pH 9 and 45 °C. Accordingly, hot springs present promising natural reservoirs for the isolation of resilient strains possessing antibacterial properties, which can be utilized in disease treatment or within the food industry.

## 1. Introduction

The rapid emergence of bacterial resistance to most commercially available antibiotics has become a pressing concern in the field of medicine. This alarming trend has led to an urgent demand for the development of novel and effective bioactive compounds with antimicrobial effects. To address this challenge, researchers and scientists have turned their attention to studying extreme environments. These unique and often harsh habitats, such as deep-sea hydrothermal vents, arctic regions, and acidic hot springs, are known to harbor extremophilic microorganisms. These microorganisms have adapted to survive in extreme conditions and possess remarkable biochemical capabilities. Exploring these extreme environments has proven to be a fruitful endeavor, as extremophilic microorganisms have demonstrated the ability to produce a diverse array of bioactive metabolites. These metabolites exhibit potent antimicrobial properties and have the potential for various commercial applications [1,2]. The discovery and isolation of these novel bioactive compounds from extremophilic microorganisms offer promising opportunities for the development of new antibiotic and antimicrobial agents. By tapping into the vast biodiversity of extreme environments, researchers can uncover previously untapped resources for combating bacterial resistance and addressing the global health challenge posed by infectious diseases. Hot springs, characterized by their extreme conditions, have garnered significant attention in recent years. These unique environments are found worldwide and can be associated with volcanic activity or exist independently [2]. It is worth noting that hot springs exhibit remarkable diversity, both in terms of their physicochemical constituents and microflora biodiversity, even among closely located ones [3]. One of the fascinating aspects of hot springs is the presence of thermophiles, microorganisms capable of thriving in these extreme environments (with temperatures ranging from 45 to 80 °C). However, our understanding of thermophiles and their biodiversity of secondary metabolites has been limited until now [4]. The exploration of hot springs has unveiled a wealth of knowledge about the adaptations and capabilities of thermophiles. These microorganisms have evolved unique mechanisms to withstand the high temperatures, extreme pH levels, and high mineral concentrations found in hot springs. Their ability to thrive in such harsh conditions has piqued the interest of researchers seeking to uncover their hidden potential [5]. One area of particular interest is the study of secondary metabolites produced by thermophiles. These metabolites are organic compounds that are not directly involved in the growth of microorganisms but often possess valuable properties. Hot springs have proven to be a rich source of diverse secondary metabolites with potential applications in various industries, including pharmaceuticals, agriculture, and biotechnology [2,6]. The discovery and characterization of these secondary metabolites from thermophiles found in hot springs hold immense promise for the development of novel bioactive compounds. These compounds may exhibit antimicrobial, anticancer, antioxidant, or other beneficial properties, making them valuable targets for drug discovery and biotechnological applications [7].

In the pursuit of novel bioactive compounds to combat drug-resistant infections, researchers have focused on isolating thermophilic bacteria from hot springs. These unique bacteria, including *Bacillus* [8,9], *Geobacillus* [10], *Brevibacillus* [11], *Pseudomonas* [12], *Paenibacillus* [13], *Aeromonas* [14], Cyanobacteria [15] and Actinobacteria species [16,17,18,19], have shown promising therapeutic potential. Hot spring ecosystems are known for their rich actinobacterial diversity [20]. Actinobacteria, a diverse group of Gram-positive bacteria with a high GC content, thrive in harsh environments and possess the ability to produce numerous therapeutic substances [21]. 

Hot springs in Iran are abundant, yet there remains a significant knowledge gap regarding their microbiological potential. Despite their potential as a valuable resource for bioactive metabolites, there have been limited publications exploring this aspect [22]. In this research, our primary objective was to isolate and characterize thermophilic bacteria with antibacterial properties from the Mahallat Hot Spring. By focusing on this unexplored hot spring, we aim to uncover novel strains of thermophilic bacteria with the ability to produce antibacterial compounds. This knowledge gap presents an exciting opportunity to explore the untapped resources within this hot spring, particularly in terms of their antibacterial-producing thermophilic bacteria.

## 2. Materials and Methods

### 2.1. Reference Strains

The reference strain of *S. aureus* (ATCC 25923) was kindly provided by Shahid Ashrafi Esfahani University (Isfahan, Iran), and the reference strain of *E. coli* (ATCC 25922) was purchased from the Iranian Biological Resource Center (IBRC), Tehran, Iran. The IBRC accession No. for *E. coli* is IBRC-M-11018.

### 2.2. Sampling of Mahallat Hot Spring

The Mahallat Hot Spring is situated near the Mahallat tourist hotel in Markazi province, specifically at coordinates 34°00′25″ N and 50°32′53″ E, which places it in the center of Iran. The sampling for this study was conducted during the autumn season. To ensure accuracy, the pH and temperature of the hot spring were directly measured on-site. Additionally, various samples, including hot spring water, sediments, sludge, and surrounding soils, were collected in zippered bags and sterile bottles using aseptic techniques. To preserve the integrity of the samples, they were promptly transferred to the laboratory while maintaining the temperature conditions of the hot spring. In the laboratory, bacteria were isolated from the collected samples for further analysis. Furthermore, the hot spring water was subjected to chemical compound analysis, pH determination, and hardness assessment by Sabzazmay Sepahan Laboratory located in Isfahan, Iran. This comprehensive approach allows for a thorough understanding of the hot spring characteristics and composition.

### 2.3. Isolation of Thermophilic Bacteria

In order to isolate thermophilic bacteria, three distinct types of growth media were employed as follows: Nutrient Agar (NA), Tryptic Soy Agar (TSA), and the solidified International Streptomyces Project medium No.2 (ISPII) containing glucose at 4 g/L; yeast extract at 4 g/L; and malt extract at 10 g/L. To isolate thermophilic bacteria from soil, sediments, and sludge, 10 g of each sample was added to 90 mL of sterile hot spring water. Following the dilution, 100 µL of the appropriate serial dilutions were transferred and evenly spread onto the agar plates. For water samples, 100 µL of the samples were directly transferred onto the plates without dilution. All the plates were then incubated for a period of 48 h at a temperature of 45 °C, maintaining aerobic conditions. After the incubation period, different colonies that had grown on the plates were carefully selected and purified using the subculture technique. Once the purity of the isolates was confirmed, a cryopreservation method was employed to ensure their long-term preservation. The isolates were carefully stored in Nutrient Broth (NB) supplemented with 20% glycerol, which is a cryoprotectant. This cryopreservation process was carried out at a temperature of −70 °C, which provides optimal conditions for maintaining the viability and integrity of the isolates. By employing this technique, the isolates can be securely stored for future experiments and analyses, ensuring their availability and usability in ongoing and future research endeavors.

### 2.4. Screening Thermophilic Bacterial Isolates for Potential Antibacterial Activity

To assess the antibacterial properties of thermophilic bacterial isolates, a cell-free supernatant was obtained from each purified isolate. Afterward, in vitro screening was performed using the agar well diffusion method on Mueller–Hinton agar [23]. The specific target bacteria chosen for this screening were the provided reference strains of *S. aureus* (Gram-positive) and *E. coli* (Gram-negative). To begin, each purified isolate was grown in the liquid form of culture media (NB, TSB (Tryptic Soy Broth), and ISPII) and incubated at a temperature of 45 °C and a speed of 150 rpm for a duration of 48 h. Following incubation, the bacterial supernatants were separated from the grown cells through centrifugation at 5000 rpm for 10 min. To ensure purity, the supernatants were then filtrated using a 0.45 µm filter membrane. For the screening process, 20 µL of the cell-free supernatant was applied to the agar wells and tested against the pathogenic reference strains (*E. coli* and *S. aureus*). The resulting inhibition zones, measured in millimeters (mm), were recorded and compared. This measurement provided an indication of the effectiveness of the isolates’ antibacterial activity [8]. Based on the diameter of the inhibition zone, the best isolates were selected for further investigation and subsequent studies. This selection process ensured that only the most promising isolates were chosen for more in-depth analysis and evaluation. The negative control consisted of a medium that was not inoculated. In order to calculate the average number of inhibition zones, three repetitions were carried out.

### 2.5. Phenotypic Characterizations

Macroscopic and microscopic morphologies, Gram reactions, and biochemical tests were examined using the methods outlined in reference textbooks [24,25].

### 2.6. Assessment of Extracellular Enzyme Production by Selected Strains

The evaluation of extracellular enzyme production by selected strains was conducted using various assays. In the amylase activity test, a 1% starch agar was utilized. The presence of amylase activity was confirmed by observing the development of a clear halo zone around the well upon the addition of a 1% freshly prepared iodine solution. This halo zone indicated the degradation of starch by the amylase enzyme. For the detection of gelatinase activity, nutrient gelatin agar (3%) was stab-inoculated with the samples. The agar plates were then incubated at 37 °C for 24 h, followed by refrigeration at 4 °C for 30 min. The presence of gelatinase activity was determined by the liquefaction of gelatin, which was considered a positive result. For the assessment of lecithinase activity, egg yolk agar was used. A streak of bacteria was made on the agar, and the formation of white precipitation around the streak indicated positive lecithinase activity. These screening methods allowed for the evaluation of amylase, gelatinase, and lecithinase activities in the samples, providing valuable insights into the enzymatic capabilities of the selected thermophilic bacterial strains [26].

### 2.7. Molecular Identification of Promising Isolates by 16S rRNA Gene Amplification and Sequence Analysis

The genomic DNA of the selected isolates was extracted from actively growing cultures that were cultivated overnight under optimal growth conditions. The extraction was performed using the boiling method, as described by Peng et al. [27]. For each bacterial isolation, the extracted DNA served as a template for amplifying the 16S rRNA gene. Universal primers, with the forward primer being 27F (5′-AGAGTTTGATCCTGGCTCAG-3′) and the reverse primer being 1492R (5′-GGTTACCTTGTTACGACTT-3′), were used for the amplification process by the PCR method. PCR reactions were performed in a 25 μL reaction volume, which included a 2X Taq DNA Polymerase Master Mix (containing 2.0 mM MgCl_2_ (Master Mix RED A190301, Ampliqon, Denmark)), with each primer at a concentration of 0.2 mM, and 1 μL of the DNA template. The remaining volume was filled with water to reach a total of 25 μL. The DNA amplification was conducted using a Thermocycler (Eppendorf, Germany) and the following conditions: initial denaturation at 94 °C for 1 min, followed by 30 cycles of denaturation at 95 °C for 30 s, annealing at 57 °C for 30 s, extension at 72 °C for 30 s, and a final extension at 72 °C for 1 min. The PCR products (5 μL) were subjected to electrophoresis in a 1% agarose gel that was prepared in a 1X TBE buffer and contained ethidium bromide. The electrophoresis was conducted for 30 min at 90 V. The DNA bands were then visualized using UV light in a gel documentation system. Following that, the purification and sequencing of the PCR products were conducted by BIOMAGIC GENE Company located in Karaj, Iran. The obtained 16S rRNA gene sequences were edited by BioEdit version 7.2.5, aligned using the CLUSTAL W multiple sequence alignment, and analyzed for homology using the ezbiocloud server (https://www.ezbiocloud.net/identify, accessed on 15 June 2022). Then, the phylogenetic trees were generated using MEGA-X software [28]. When utilizing the neighbor-joining method, distances in the trees were calculated using Kimura’s two-parameter model. Furthermore, 1000 resampling iterations were performed to determine the bootstrap values [29]. The nucleotide sequences of the amplified 16S rRNA genes from the chosen isolates in this research were recorded in the GenBank nucleotide sequence databases with distinct accession numbers.

### 2.8. Growth Kinetics of Antibacterial-Producing Strains

In this experiment, 500 mL flasks were prepared, each containing 200 mL of autoclaved NB for the isolates Kh2 and Kh5 and the ISPII medium for the isolate Kh3. To initiate the cultures, 20% (*v*/*v*) of three isolates’ precultures with an Optical Density (OD) of 0.1 at A600 nm were inoculated into the respective flasks. The flasks were then incubated at 45 °C with continuous shaking at 150 rpm. The growth of bacteria was monitored by measuring the OD_600_ every two hours for the duration of 72 h for the isolates Kh2 and Kh5 and 9 days for the isolate Kh3. To assess the antibacterial activity, the cell-free supernatant was collected at 24 h intervals and tested against *S. aureus* and *E. coli* using the inhibition-zone measurement method described earlier. All experiments were performed in triplicates to ensure the reliability of the results.

### 2.9. Physiological Characterizations

The growth temperature range was determined by incubating isolates at temperatures ranging from 10 to 80 °C with intervals of 10 °C. The growth’s dependence on pH was tested by varying the pH from 5.0 to 10.0. The impact of NaCl on the growth of thermophilic bacteria was studied in an NB medium with varying concentrations of NaCl from 0 to 20% (*w*/*v*) NaCl [30]. All experiments were conducted in triplicate.

### 2.10. Heat Stability of Cell-Free Supernatant

To assess the thermostability of the cell-free supernatant from the selected strains, a series of incubation experiments was conducted. The supernatant samples were subjected to different temperatures, specifically 40, 50, 60, 70, 75, and 80 °C, for 30 min. For each experiment, 1 mL of the cell-free supernatant was heated to the desired temperature and subsequently cooled to room temperature. The cooled samples were then tested for inhibitory activity against two specific reference strains, namely *S. aureus* and *E. coli*, using the agar well diffusion assay. This assay involved creating wells in an agar plate and adding the cooled, heat-treated supernatant samples into these wells. After measuring the diameter of the halo zones, the maximum values for the bacteria were considered as 100%, with any changes at higher temperatures compared to this value reported as a percentage. The inhibitory activity of the supernatant against the bacteria was determined by observing the extent of growth inhibition around the wells. By conducting these experiments, we aimed to evaluate the impact of different temperatures on the stability and inhibitory activity of the cell-free supernatant, providing valuable insights into its potential applications in combating bacterial growth.

## 3. Results

### 3.1. Isolation and Screening of Thermophilic Bacteria with Antibacterial Activity

Mahallat Hot Spring, located in the central region of Iran within the Markazi province, is renowned among locals for its exceptional therapeutic properties. During the sampling process, the water from the hot spring was found to have a temperature of 45 °C and a pH level of 6.9. In order to gain a comprehensive understanding of the water composition, a thorough analysis of the water sample was conducted. This analysis revealed the presence of several cationic and anionic inorganic ions, including calcium, magnesium, sodium, potassium, chloride, bicarbonate, sulfate, and nitrate. To provide a more detailed overview, Table 1 presents the concentrations of these ions and other water-specific properties.

From various collected hot spring samples, 19 different colonies were isolated and purified. To evaluate the antibacterial activity of the pure bacterial isolates, a screening process was conducted using the reference strains of *S. aureus* and *E. coli*. Out of the total 19 bacterial isolates, it was found that only 8 of them, accounting for 42% of the isolates, exhibited antibacterial activity against both tested pathogens. It was revealed that three isolates demonstrated the highest antibacterial inhibition zones, as depicted in Table 2. These three isolates were selected for further investigation due to their exceptional antibacterial properties.

### 3.2. Phenotypic Characterizations of Selected Thermophilic Bacterial Isolates

Microscopic and macroscopic morphology and some biochemical characteristics of three selected thermophilic bacterial isolates with the highest antibacterial activity were investigated based on routine microbiological methods. The selected isolates designating Kh2 and Kh5 belonged to the genus *Bacillus*, while the isolate Kh3 showed similarity to the genus *Saccharomonospora*. Phenotypic characterizations of these three isolates are shown in Table 3 and Figure 1. 

### 3.3. Molecular Identification and Phylogenetic Analysis of Selected Thermophilic Bacterial Isolates

Based on their antibacterial activity, three highly promising isolates, Kh5, Kh2, and Kh3, were chosen to be molecularly identified through 16S rRNA gene amplification and sequencing. To further understand their taxonomic classification, the sequenced 16S rRNA genes were compared with the EzBioCloud database (https://www.ezbiocloud.net/identify, accessed on 15 June 2022). This analysis revealed that Kh2 and Kh5 belonged to the genus *Bacillus*, with a remarkable similarity of 99.79% and 99.93% to *Bacillus halotolerans*, respectively. On the other hand, the 16S rRNA gene sequence of Kh3 showed a 99.78% similarity to the species *S. azurea* from the family *Pseudonocardiaceae*. To ensure the accessibility and traceability of these valuable findings, the 16S rRNA gene sequences of these three strains (Kh2, Kh3, and Kh5) were deposited in the GenBank sequence database, which can be accessed at https://www.ncbi.nlm.nih.gov/, 7 October 2022. Each strain was assigned a unique GenBank accession number for their respective 16S rRNA gene sequences. Specifically, the accession numbers for Kh2, Kh3, and Kh5 were OP564888, OP564889, and OP564890, respectively. Additionally, phylogenetic trees depicting the relationships of these strains with the type strains of relative species were constructed using the neighbor-joining method in Mega-X software (Figure 2 and Figure 3).

### 3.4. Growth Kinetics of Selected Thermophilic Strains and Their Antibacterial Activity

Following the identification of the selected isolates, such as *Bacillus* (Kh2 and Kh5 strains) and *Saccharomonospora* (Kh3 strain), the physiology of their growth in respective culture media and simultaneously their antibacterial activity were examined. The incubation temperature was set to 45 °C, and the duration of incubation ranged from 1 to 9 days. The study of the growth pattern of two *Bacillus* sp. strains, Kh2 and Kh5, was checked in the TSB medium. The analysis of growth curves revealed that both strains had a lag phase followed by a log phase. Strains Kh2 and Kh5 entered the exponential phase approximately 10 h after the initial inoculation. The stationary phase was determined at 18 h and 22 h for strains Kh2 and Kh5, respectively (Figure 4a,b). Similarly, the growth pattern of strain Kh3, identified as *S. azurea*, was examined in the ISPII medium. After 24 h from the initial inoculation, this strain passed in the log phase, and its stationary phase was detected after 6 days (Figure 4c).

Based on the inhibitory zone, the antibacterial activity of the crude supernatants of three strains was determined during the study of their growth patterns. For two *Bacillus* sp. strains (Kh2 and Kh5), the antibacterial activity became detectable after 24 h of growth and reached its maximum after 48 h. No further activity was detected after 48 h, indicating the entry of the bacterial growth cycle into the death phase (Figure 4a,b). The crude supernatant of *S. azurea* strain Kh3 exhibited measurable antibacterial activity after 24 h, similar to the *Bacillus* sp. strains. However, its maximum antibacterial activity was detected after 4 days. Subsequently, no further activity was observed after 120 h of initial inoculation (Figure 4c).

### 3.5. Physiological Characterizations of Selected Thermophilic Strains

In order to assess the physiological characteristics of the promising thermophilic strains (Kh2, Kh3, and Kh5), an extensive investigation was conducted to determine their cardinal temperature and pH values, as well as their tolerance to various NaCl concentrations. The temperature range for the strains was systematically studied, covering a wide spectrum from 10 °C to 75 °C. Through particular observation, it was found that the optimal temperature for growth varied amongst the strains. Specifically, Kh2 and Kh5 exhibited the highest growth rate at 40 °C, while Kh3 reached maximum growth at a temperature of 45 °C. Notably, no substantial differences in growth were observed within the temperature range of 20 to 40 °C for the *Bacillus* sp. strains. Moreover, the highest temperature that the *Bacillus* sp. strains (Kh2 and Kh5) and *S. azurea* strain Kh3 could tolerate was 60, 75, and 70 °C, respectively. Furthermore, no growth was observed for all strains at 10 °C (Figure 5a). After determining the optimal growth temperature for all strains, the suitable pH range for their growth was conducted. The obtained results indicated that *Bacillus* sp. strains (Kh2 and Kh5) preferred a wide range of pHs from 5 to 8 for their maximum growth. However, for the strain Kh3, it was revealed that the alkaline condition (pH 9) rather than acidic conditions was more favorable for its growth at 45 °C. In total, no considerable growth was detected at pH 9 and 10 for the other two strains (Figure 5b).

Subsequently, the influence of salt concentration on the growth of these thermophilic strains (Kh2, Kh3, and Kh5) was examined. Surprisingly, none of the strains required salt for growth, as they exhibited their highest growth rate in salt-free media. However, it was noted that salt concentrations of up to 5% had minimal effects on the strains Kh2 and Kh5 growth. The growth of Kh2 in the presence of 7.5% (*w*/*v*) NaCl was decreased, while strain Kh5 maintained normal growth. Both strains, however, showed tolerance to 10% (*w*/*v*) NaCl but showed no considerable growth in the presence of 15% and 20% NaCl (*w*/*v*). Strain Kh3, similar to *Bacillus* sp. strains, was found to have a tolerance to NaCl of up to 10%, with no significant growth in the presence of 15% and 20% NaCl (*w*/*v*) (Figure 5c).

### 3.6. Heat Stability of the Cell-Free Supernatant of Selected Thermophilic Strains

During heat stability testing, it was detected that the antibacterial compounds derived from all strains remained stable within a temperature range of 30–75 °C after 30 min of incubation (Figure 6). Interestingly, the size of the inhibition zone had only a minor decrease when supernatants were heated up to 75 °C for 30 min. There was no detectable antibacterial activity following exposure to a higher temperature (80 °C).

These findings suggest that the antibacterial compounds produced by thermophilic strains exhibited heat resistance, which enhances their potential applications across various fields.

## 4. Discussion

Extreme environments, such as terrestrial hot springs, contain a wide range of diverse and resistant microbial communities that differ in taxonomy and functionality. Hot springs have been comprehensively studied by researchers worldwide as a potential source of thermophilic bacteria. This type of extremophilic bacteria is considered a promising candidate for producing novel functional metabolites to address current global challenges, such as antibiotic resistance [31]. According to recent reports, there is emergent concern that antibiotic resistance could pose a greater threat to human life in the future, potentially surpassing the impact of cancer. It is anticipated that, by the year 2050, up to 10 million lives could be lost due to the rise of bacterial resistance to antibiotics. This frightening prediction highlights the urgency to address this issue and take appropriate actions to contest the growing threat of antibiotic resistance [32]. Hence, there is currently a rising demand for the discovery of ground-breaking and powerful antimicrobial compounds sourced from unexplored and extreme natural environments like hot springs. While numerous hot springs have been documented in Iran, only a limited number of studies have been conducted on their microbial communities [33].

In this study, a total of 19 thermophilic bacteria were successfully isolated from the Mahallat Hot Spring in Iran, and out of these isolates, three exhibited significant antibacterial activity. Located in a relatively unexplored region, this hot spring is renowned for its unique characteristics, including an exceptionally high level of natural radiation. Despite its rich biodiversity, it remains relatively untapped in terms of scientific exploration. The maximum radiation concentration of 226Ra in the soil is recorded at 13,000 Bq/kg, while for water, it is 130 Bq/L. The temperature measurement of the spring was recorded at 45 °C, which closely aligns with the temperature (46 °C) reported by Beitollahi et al. [34]. Polyphasic taxonomy analyses revealed that three promising isolates were categorized as aerobic, Gram-positive, and spore-forming bacteria. Two of them were classified as the genus *Bacillus*, while another isolate was identified as the genus *Saccharomonospora* and the species *S. azurea*. They grew normally on synthetic culture media, including TSA, NA, and ISPII, and did not show any compulsory requirement for the growth factors, such as vitamins. Comparable results were attained by Aissaoui et al. [35]. Strain Kh3, which was defined as a thermophile, showed its highest growth at 45 °C, and no growth was detected higher than 75 °C. In total, they could able to grow within a temperature range from 20 to 75 °C [36]. *Bacillus* sp. strains (Kh2 and Kh5) grew optimally at a pH range from 5.0 to 8.0. But the strain Kh3 had its maximum growth at pH 9. Their growth was also checked for NaCl contents between 0 and 20%, with an optimum growth at 0%. Also, all strains were able to tolerate NaCl up to 10%.; therefore, they were halotolerant, as defined by Kushner [37]. According to the polyphasic taxonomic studies, accurately identifying isolates belonging to the genus *Bacillus* solely based on their phenotypic characteristics and 16S rRNA gene sequencing is not forthright. This complexity arises from the significant evolutionary heterogeneity within the *Bacillus* genus, making species identification a challenging task. Therefore, a comprehensive approach utilizing multiple taxonomic markers becomes essential for accurate classification and identification [38]. As a result, the *Bacillus* strains studied in this research were not assigned to any specific species.

From a biotechnological point of view, thermophiles can produce various distinctive bioactive materials that may not be widely studied, and the available research mostly focuses on thermozymes and their industrial applications [35]. In this study, we specifically targeted the antibacterial activity of the identified promising strains, as antibiotic resistance has emerged as a serious global health concern. The genera *Bacillus* and *Actinomycetes* are distinguished for being considerable sources of bioactive compounds and various secondary metabolites. However, a small amount of research has been conducted on the potential of their extremophilic strains [39]. In this study, the strain Kh3 was identified as *S. azurea*, which is a rare species from *Actinomycetes*. This species is a fascinating species that has garnered attention due to its potential in the production of antibacterial activity, particularly against Gram-positive bacteria that have developed resistance to multiple drugs [40,41]. Feiszt et al. [42] conducted a re-evaluation of the antibacterial activity of *S. azurea* against various bacterial pathogens in vitro. Their findings showcased the efficacy of pyrimycin, an antibacterial substance present in *S. azurea*, in battling MRSA compared to the tested antibiotics. However, it is important to note that pyrimycin did not demonstrate effectiveness against any Gram-negative bacteria. According to the results we obtained, it is demonstrated that the Kh3 strain exhibits antibacterial activity against both Gram-negative and Gram-positive bacteria. The production of other bioactive metabolites with antimicrobial and anticancer properties has also been reported in the various species of *Saccharomonospora*. Interestingly, most previously investigated strains were obtained from soil samples and were found to be mesophilic, exhibiting optimal growth at 28 °C. These strains displayed a growth range spanning from 24 °C to 40 °C [43]. In our research, we successfully isolated the *S. azurea* strain Kh3 from the mud of Mahallat Hot Spring, Iran. This particular strain demonstrated impressive growth capabilities, thriving at elevated temperatures of up to 75 °C and alkaline conditions with a pH of 9. Additionally, it exhibited a noteworthy adaptability to a wide range of temperatures varying from 20 °C to 75 °C and a pH range of 6 to 10. This finding highlights the resilience and versatility of the thermoalkaliphilic *S. azurea* strain Kh3, making it an exciting prospect for further scientific exploration and potential applications. Overall, thermoalkaliphilic bacteria usually exhibit a preference for anaerobic growth conditions, whereas the occurrence of aerobic variants is relatively scarce [30]. Regarding the isolated region’s type and the presence of harsh environmental factors like radiation and high temperature, *S. azurea* strain Kh3 can exhibit multiple mechanisms to cope with these conditions, making it a polyextremophile. Polyextremophiles are valuable sources of highly resilient bioactive materials with significant potential. The supernatant of the *S. azurea* strain Kh3, which was incubated at a temperature range of 30–75 °C for 30 min, demonstrated remarkable thermostability. It displayed significant antibacterial activity against reference pathogens. Furthermore, strain Kh3 revealed enzymatic activity, indicating its potential for effectively breaking down various materials. Exploiting this capability, we developed a cost-effective culture medium to cultivate this bacterium and produce antimicrobial substances on a large scale, making it suitable for industrial applications.

The genus *Bacillus* is known for its ability to produce a wide range of substances, including antibiotics, antimicrobial peptides, and enzymes. In this study, both strains (Kh2 and Kh5) of the genus *Bacillus* were examined, and it was found that they could thrive in various temperature ranges, with the optimal growth occurring at 40 °C. These strains were also found to tolerate a wide pH range of 5 to 9 and high NaCl concentrations of up to 15%. One remarkable characteristic of both Kh2 and Kh5 strains was their significant antibacterial activity against both Gram-positive and Gram-negative bacteria. This suggests their potential to produce active metabolites with antimicrobial properties. The genus *Bacillus* has been extensively studied for its biocontrol abilities, particularly in terms of its antimicrobial and anti-nematode effects [44]. For instance, research has shown that *B. halotolerans* can produce a hemolytic lipopeptide that exhibits antimicrobial properties against *S. aureus* and *S. cerevisiae* [45]. Overall, the findings of this work highlight the promising capabilities of both Kh2 and Kh5 strains of the genus *Bacillus*, paving the way for further exploration and utilization of their microbial active metabolites in various applications.

## 5. Conclusions

In this study, three promising thermoalkaliphilic and thermotolerant bacterial strains revealing antibacterial activity against reference pathogens were isolated from the Mahallat Hot Spring in Iran. Their potent antibacterial properties have paved the way for possible medical applications in the field of infectious diseases. Moreover, this research sheds light on the enormous potential of this hot spring and its diverse bacterial community. By delving into this unique environment with harsh conditions, a deeper understanding of the interplay between antibacterial properties and the remarkable adaptations of microorganisms within this extreme ecosystem can be gained.

## Figures and Tables

**Figure 1 microorganisms-12-00834-f001:**
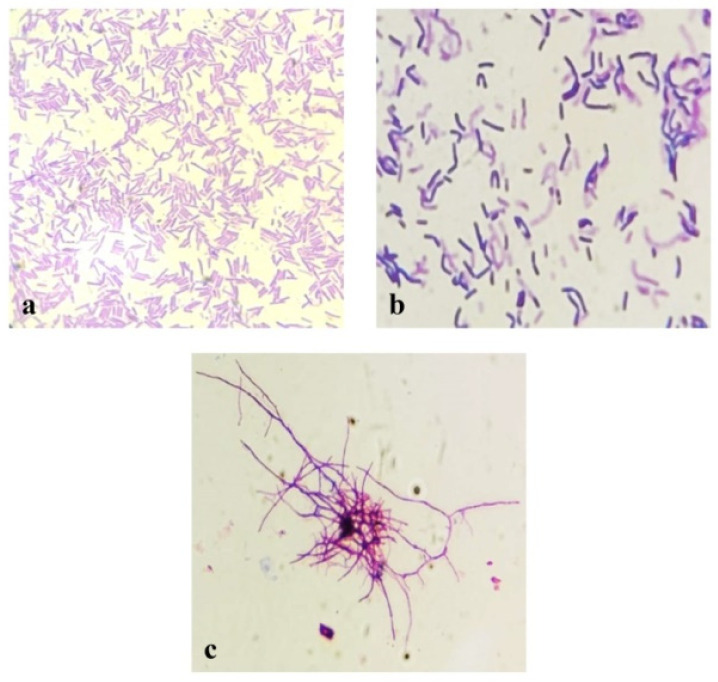
Microscopic morphology of three designated strains on TSA (24–48 at 45 °C): (**a**) Kh2; (**b**) Kh5; and (**c**) Kh3. (Scale bar: 20 µm).

**Figure 2 microorganisms-12-00834-f002:**
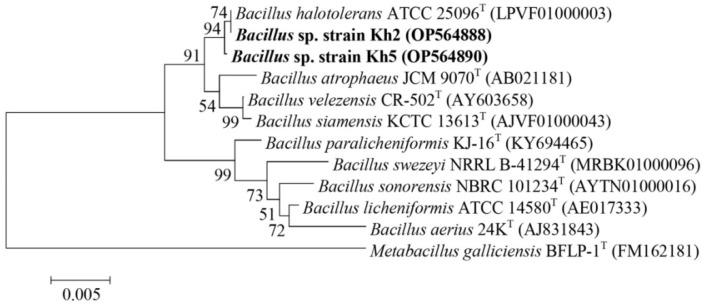
Neighbor-joining phylogenetic tree based on 16S rRNA gene sequences for Kh2 and Kh5 strains, revealing the positions of these strains and other related species. Highlighted in bold are the positions of the strains Kh2 and Kh5 in the phylogenetic tree. GenBank accession numbers are shown in parentheses. Bootstrap values (%) are based on 1000 replicates. Bar 0.005 substitutions per nucleotide position. *Metabacillus galliciensis* BFLP-1T was chosen as an out-group.

**Figure 3 microorganisms-12-00834-f003:**
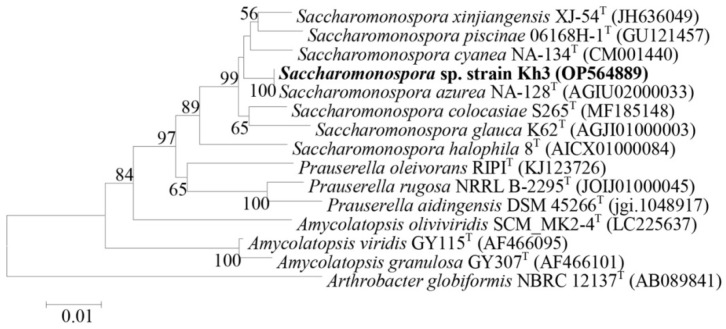
Neighbor-joining phylogenetic tree based on 16S rRNA gene sequences for the Kh3 strain, revealing the position of this strain (highlighted in bold) and other related species. GenBank accession numbers are shown in parentheses. Bootstrap values (%) are based on 1000 replicates. Bar 0.01 substitutions per nucleotide position. *Arthrobacter globiformis* NBRC 12137T was chosen as an out-group.

**Figure 4 microorganisms-12-00834-f004:**
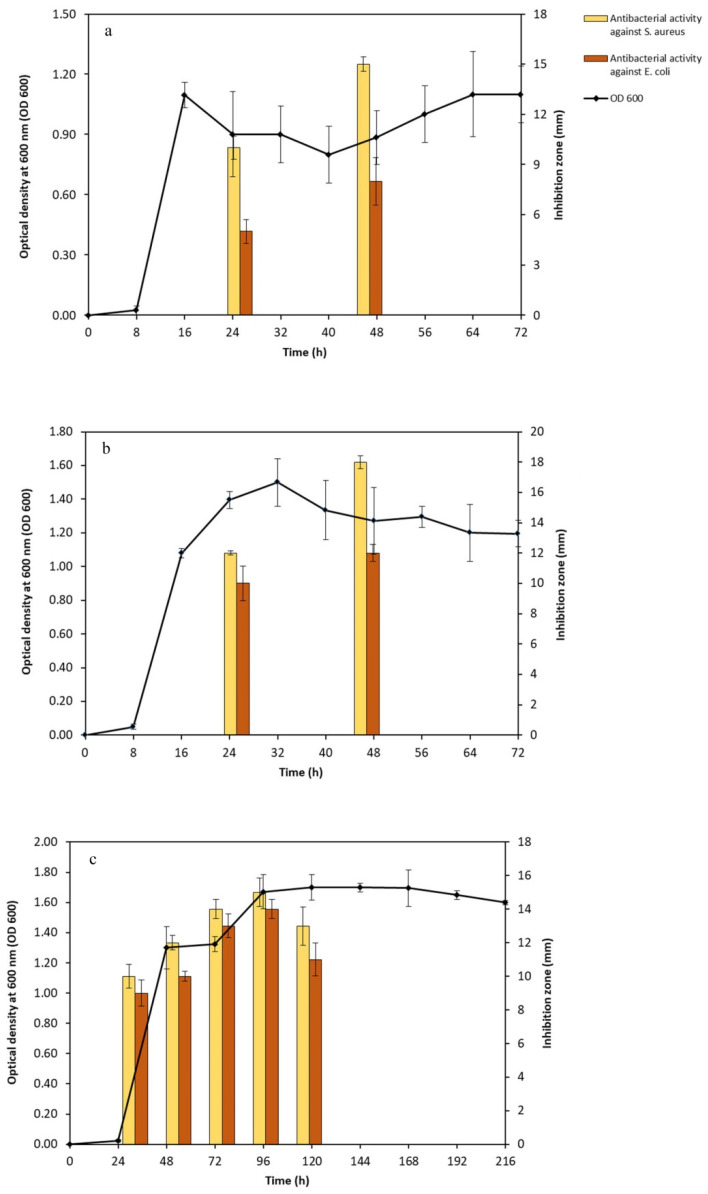
Growth curve and antibacterial production in three selected thermophilic strains; (**a**) strain Kh2; (**b**) strain Kh5; and (**c**) strain Kh3. Antibacterial activity (inhibition zone in mm) at different growth phases was checked. All strains showed the best antibacterial activity at the stationary phase. The inhibition zones developed were more prominent against *S. aureus* vs. *E. coli*.

**Figure 5 microorganisms-12-00834-f005:**
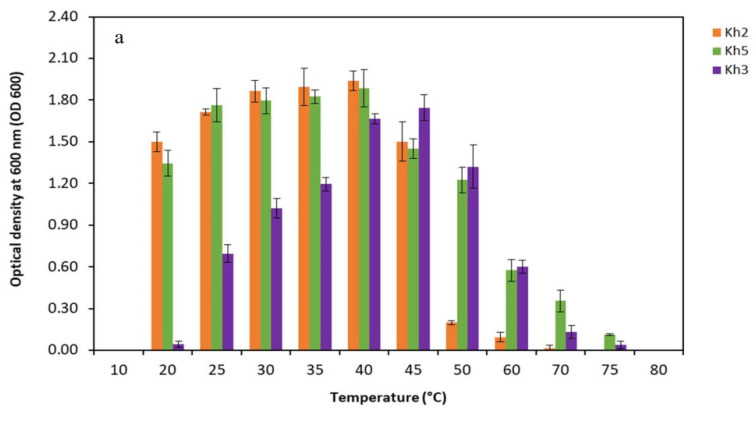
Physiological characterizations of selected thermophilic strains. Determination of (**a**) cardinal temperatures; (**b**) cardinal pH values at the optimal temperature of each strain; and (**c**) tolerance to NaCl concentrations for three selected strains at the optimal temperature of each strain (Kh2, Kh5, and Kh3).

**Figure 6 microorganisms-12-00834-f006:**
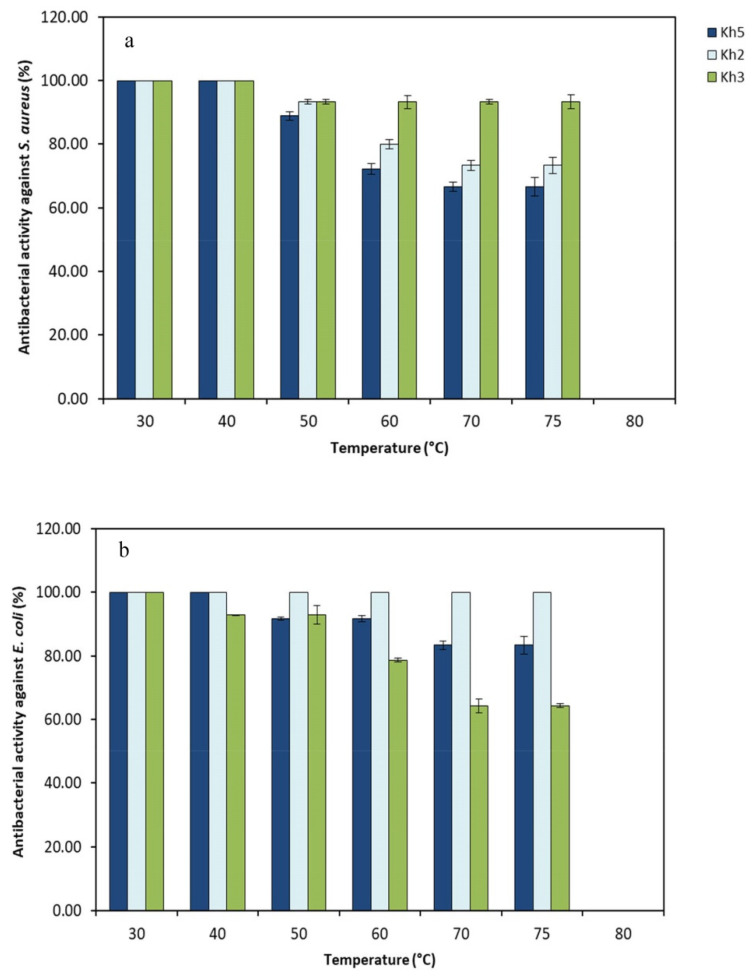
Antibacterial activity (%) of three selected strains (Kh2, Kh3, and Kh5) against (**a**) *S. aureus* and (**b**) *E. coli* after heat treatments in various temperatures.

**Table 1 microorganisms-12-00834-t001:** Chemical analysis of Mahallat Hot Spring water performed by Sabzazmay Sepahan Laboratory (Isfahan, Iran).

Parameter	Amount	Unit
Electrical conductivity (EC)	1.8	dS/m
Total Dissolved Solids (TDSs)	1770	mg/L
Total Suspended Solids (TSSs)	≤50	mg/L
[SO_4_^2−^]	960	mg/L
[HCO_3_^−^]	244	mg/L
[CO_3_^2−^]	0.0	mg/L
[Cl^−^]	49.7	mg/L
[NO_3_^−^]	0.14	mg/L
[Ca^2+^]	360	mg/L
[Na^+^]	87.4	mg/L
[Mg^2+^]	57.6	mg/L
[K^+^]	5.8	mg/L
Total Hardness (TH)	1140	P.P.M (France Degree)
pH	6.95	-

**Table 2 microorganisms-12-00834-t002:** Antibacterial activity of the selected thermophilic bacterial isolates.

Isolate Designation	Isolation Source	Zone of Inhibition (mm)	
		*S. aureus*	*E. coli*
Kh5	Hot spring sludge	18	12
Kh2	Soil	15	8
Kh3	Hot spring sludge	12	10

**Table 3 microorganisms-12-00834-t003:** Various phenotypic characteristics of the designated strains on TSA (24–48 at 45 °C): 1, Kh2; 2, Kh5; and 3, Kh3. ND, not determined; −, negative; +, positive; and ±, variable.

Characteristics	Kh2	Kh5	Kh3
Colony morphology	Smooth, circular with undulate margin, wrinkled, opaque, and slightly raised	Rough, circular with undulate margin, opaque, and flat	Rough, circular with undulate margin, opaque, raised, and adhered to the surface of agar
Colony color	Creamy	Creamy	White-powdery surface with creamy color for substrate mycelia
Pigment	−	−	+
Pigment color	−	−	Non-diffusible green
Colony size (mm)	2 mm	2 mm	1 mm
Cell morphology	Rod-shape occurring singly and palisade form	Rod-shape occurring singly, in pairs, and sometimes palisade form	Filamentous, Branches of mycelia fragmented into rod or coccoid-shaped forms
Cell size (µm)	ND	ND	<1 µm in diameter
Gram staining	+	+	+
Reaction with KOH 3%	−	−	−
Motility	+	+	−
Spore formation	+	+	+
Spore morphology	Spherical	Spherical	Spherical
Spore position in the cell	Central to subterminal	Central to subterminal	Released from cells
Aerobic growth	+	+	+
Anaerobic growth	+	+	−
Growth in O/F medium	ND	ND	±
Acid produced from the following:			
D-Arabinose	ND	ND	−
D-Xylose	ND	ND	−
D-Galactose	ND	ND	−
D-Glucose	+	+	+
D-Mannose	+	+	−
Cellobiose	ND	ND	−
Sucrose	−	−	+
Glycerol	ND	ND	ND
Methyl Red/Voges Proskauer (MR/VP) reaction	±	±	±
Gas formation from carbohydrates	ND	ND	−
Indole formation	+	+	+
H_2_S formation	−	−	−
Nitrate reduction	+	+	+
Catalase activity	+	+	+
Oxidase activity	−	−	−
Amylase activity	−	−	+
Gelatinase activity	−	−	+
Lecithinase activity	+	+	+

## Data Availability

Data are contained within the article.

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
