# Peer review of "Unveiling Antibacterial Potential and Physiological Characteristics of Thermophilic Bacteria Isolated from a Hot Spring in Iran"

_microorganisms, 2024, doi:10.3390/microorganisms12040834_

Round 1
Reviewer 1 Report
Comments and Suggestions for Authors
I have gone through the manuscript entitled “Unveiling Antibacterial Potential and Physiological Characteristics of Thermophilic Bacteria Isolated from a Hot Spring in Iran”. In this study, authors have isolated thermophilic bacteria from Mahallat hot spring, Iran, finding some isolates displaying remarkable antibacterial activity. The manuscript is properly written. However, I regret to inform that I believe that the article does not meet the standards for publication in the journal Microorganisms. Therefore, I recommend rejecting the article.
1. The 16S rRNA gene amplification and sequence analysis do not offer accurate classification and identification.
2. The presented data is preliminary, leading to a lack of substantial discussion. The discussion section mainly repeats the results and provides only a brief literature review and superficial discussion on the potential of the isolated S. azurea strain Kh3.
3. Overall, the content lacks significance, scientific soundness, and interest.
Comments on the Quality of English LanguageThe overall quality of the English language in the manuscript is acceptable; however, minor editing is recommended to enhance clarity and coherence. Some sentences could benefit from a smoother flow and better organization of ideas.
Author Response
Authors’ Responses to Reviewer 1:
Comments and Suggestions for Authors
I have gone through the manuscript entitled “Unveiling Antibacterial Potential and Physiological Characteristics of Thermophilic Bacteria Isolated from a Hot Spring in Iran”. In this study, authors have isolated thermophilic bacteria from Mahallat hot spring, Iran, finding some isolates displaying remarkable antibacterial activity. The manuscript is properly written. However, I regret to inform that I believe that the article does not meet the standards for publication in the journal Microorganisms. Therefore, I recommend rejecting the article.
- The 16S rRNA gene amplification and sequence analysis do not offer accurate classification and identification.
- Thank you for reviewing our manuscript and providing valuable feedback. We greatly appreciate your insights and constructive criticism. We understand your concerns regarding the accuracy of the 16S rRNA gene amplification and sequence analysis for classification and identification in microbiology research. While precise taxonomic classification is indeed important in microbiology, the primary objective of our research was to identify novel antibacterial agents from an extreme environment. Through our analysis of the 16S rRNA gene, we determined that the strains under study did not belong to a novel taxon. Given our research's financial constraints, we made the decision not to pursue further genomic analyses.
- The presented data is preliminary, leading to a lack of substantial discussion. The discussion section mainly repeats the results and provides only a brief literature review and superficial discussion on the potential of the isolated azureastrain Kh3.
- Thank you for your insightful feedback on the manuscript. We appreciate your thorough evaluation of the discussion section and acknowledge the need for a more substantial analysis of the data and a deeper exploration of the literature. We tried to revise the manuscript to address these concerns and ensure that the manuscript meets the publication standards of the journal Microorganisms.
- Overall, the content lacks significance, scientific soundness, and interest.
- Thank you for your feedback on the manuscript. We appreciate your assessment of the content and acknowledge your concerns regarding its significance, scientific soundness, and interest. We carefully reviewed and revised the manuscript to address these issues and enhance the overall quality of the research.
Comments on the Quality of English Language
The overall quality of the English language in the manuscript is acceptable; however, minor editing is recommended to enhance clarity and coherence. Some sentences could benefit from a smoother flow and better organization of ideas.
- Thank you for your comment. We revised the manuscript to be sure about the quality of English language.

Reviewer 2 Report
Comments and Suggestions for Authors
The manuscript adequately describes the isolation of thermolytic bacteria and the antimicrobial activity detected in the supernatants of their cultures. However, there are certain major uncertainties that must be resolved. 1.- The activity is detected in direct supernatants. Have the biomasses been tested to confirm that they are only extracellular compounds? 2.- Possibly, the responsible for the activity are siderophores. It would be very important to scale up and extract the cultures to elucidate the antibacterial substances. Or, at least, de novo sequencing to perform genome mining (i.e. antiSMASH) to identify biosynthetic gene clusters of bioactive metabolites or siderophores (as NRPS or lassopeptides). 3.- In the case of Bacillus strains, there are only 2 sampling (24 and 48h), it is desirable to obtain complete kinetics (from 0h to 96h) to verify under what conditions the maximum activity occurs.
Finally, the strains Kh2 and Kh5 are described such as "different" strains. What is the molecular argument for this affirmation? The 16S rRNA similitude is not relevant to distinguished strains. A DNA fingerprinting (i.e. Repetitive PCR) could help to discriminate strains.
Author Response
Authors’ Responses to Reviewer 2:
Comments and Suggestions for Authors
The manuscript adequately describes the isolation of thermolytic bacteria and the antimicrobial activity detected in the supernatants of their cultures. However, there are certain major uncertainties that must be resolved.
- The activity is detected in direct supernatants. Have the biomasses been tested to confirm that they are only extracellular compounds?
- Thank you for your feedback. In response to your query, we initially observed antibacterial activity around the strains' colonies, indicating a growth inhibition zone when cultured with pathogenic strains. While traditional methods involve extracting bioactive components from dried biomass for testing against pathogenic bacteria, we focused on evaluating fresh culture and supernatant in this research. We appreciate your suggestion and will consider incorporating additional experiments to extract and test bioactive compounds from the biomass in future studies for a more comprehensive analysis.
- Possibly, the responsible for the activity are siderophores. It would be very important to scale up and extract the cultures to elucidate the antibacterial substances. Or, at least, de novo sequencing to perform genome mining (i.e. antiSMASH) to identify biosynthetic gene clusters of bioactive metabolites or siderophores (as NRPS or lassopeptides).
- Thank you for your valuable feedback. As we continue our investigation into the antibacterial activities of thermophiles, our research will progress to include a metagenomic study focusing on identifying the biosynthetic genes responsible for the production of antibacterial compounds. This next step in our research aims to provide a deeper understanding of the mechanisms underlying the observed antibacterial activity.
- In the case of Bacillus strains, there are only 2 sampling (24 and 48h), it is desirable to obtain complete kinetics (from 0h to 96h) to verify under what conditions the maximum activity occurs.
- Thank you for your input. We would like to inform that we have already investigated time points at 0h, 72h, and 96h. Regrettably, our findings did not reveal significant antibacterial activities during these time intervals.
- Finally, the strains Kh2 and Kh5 are described such as "different" strains. What is the molecular argument for this affirmation? The 16S rRNA similitude is not relevant to distinguished strains. A DNA fingerprinting (i.e. Repetitive PCR) could help to discriminate strains.
In our research, we have primarily focused on phenotypic characterization of the Bacillus strains, which has allowed us to distinguish them based on physiological, microscopic, and macroscopic features. However, to further enhance the identification process, we are planning to incorporate Multilocus Sequence Analysis (MLSA) for a more precise and comprehensive analysis of the strains.

Reviewer 3 Report
Comments and Suggestions for Authors
The article by Rafiee et al. is devoted to the isolation and characterization of bacterial strains from the Mahallat hot spring in Iran that have antibacterial potential. Of the 30 isolated strains, three were useful for this purpose. Interestingly, the isolated strains had different temperature optimums for growth and responded differently to environmental pH and NaCl concentration. This indicates the inexhaustible potential of Iranian hot springs for the search for various thermotolerant bacteria, the study of which can significantly expand the known diversity.
It has been shown that the antibacterial activity of isolated strains against E. coli and S. aureus is maintained up to a temperature of 75oC; this interesting fact can be used in pharmacology and medicine.
The methods are described in detail. All references in the Introduction and Discussion are relevant to this work.
Minor revisions:
In the caption to Figure 5, you must first give the general name of the Figure, and then write what they mean: a) b) c)
Reference:
No title in link 1.
In magazine titles, all words should be capitalized.
Author Response
Authors’ Responses to Reviewer 4:
Comments and Suggestions for Authors
The article by Rafiee et al. is devoted to the isolation and characterization of bacterial strains from the Mahallat hot spring in Iran that have antibacterial potential. Of the 30 isolated strains, three were useful for this purpose. Interestingly, the isolated strains had different temperature optimums for growth and responded differently to environmental pH and NaCl concentration. This indicates the inexhaustible potential of Iranian hot springs for the search for various thermotolerant bacteria, the study of which can significantly expand the known diversity. It has been shown that the antibacterial activity of isolated strains against E. coli and S. aureus is maintained up to a temperature of 75 °C; this interesting fact can be used in pharmacology and medicine. The methods are described in detail. All references in the Introduction and Discussion are relevant to this work.
- Thank you for reviewing our manuscript and providing valuable feedback
Minor revisions:
In the caption to Figure 5, you must first give the general name of the Figure, and then write what they mean: a) b) c)
- Thank you for your suggestion. It was revised as below and highlighted in yellow in the revised version of the manuscript.
“Figure 5. Physiological Characterizations of selected thermophilic strains. Determination of a) cardinal temperatures; b) cardinal pH values at optimal temperature of each strains; and c) tolerance to NaCl concentrations for three selected strains at optimal temperature of each strains (Kh2, Kh5, and Kh3)”.
Reference:
No title in link 1.
- Thank you for the consideration. It was corrected and highlighted in yellow (page 17).
“Thakur, N., Singh, S. P. & Zhang, C. Microorganisms under extreme environments and their applications. Current Research in Microbial Sciences Vol. 3, 100141 (Elsevier, 2022). https://doi.org/10.1016/j.crmicr.2022.100141
“
In magazine titles, all words should be capitalized.
- Thanks for your comment. We have checked the title and no words were written with small letter.

Reviewer 4 Report
Comments and Suggestions for Authors
The authors investigated the antibacterial potential of three thermophilic bacteria (two Bacillus sp. strains and S. azurea strain) isolated from the Mahallat hot spring in Iran. All three isolates revealed antibacterial activity against control pathogenic strains of S. aureus (Gram-positive) and E. coli (Gram-negative). Their potential antibacterial properties may be useful for possible medical applications in the field of infectious diseases.
I do find this work very interesting and valuable.
However, there are some issues which should be addressed before publishing.
Specific comments:
The Title, Lines 2-3 – “Characteristics”, not “Character-istics”.
Figure 6 – The title of the Y axis: How did you calculate this value? Please add this in the Methods.
Table 2: Columns 3,4 – Mean or maximum values? Please adjust it to the values in Fig. 4.
Line 134 – “TSB”? Please decode it.
Lines 208-209 – “temperatures ranging from 30 to 80° C”. I suppose “from 10 to 80 °C”. See Fig. 5a.
Line 212 – Is it correct: “%(w/v) NaCl”?
Lines 267 – “through 16S rRNA” instead of “through16S rRNA”.
Lines 301,306 – “lag phase” and “log phase”. Please correct it.
Lines 333-334 – “temperature that Bacillus sp. strains (Kh2 and Kh5) and S. azurea strain Kh3 could tolerate was 75 and 60 °C, respectively”. May be, 60 and 75 °C for Kh2 and Kh5, respectively, and 70 °C for Kh3? See Fig. 5a.
Line 340 – “no considerable growth was detected at pH 9 and 10 for all three strains”. A considerable growth of Kh3 was detected at pH 9 (see Fig. 5b and Line 339). Please correct it.
Lines 345, 347, 348 and 350 – What did you mean: weight (% w/w) or mass (% w/v) concentration of NaCl solution? In Methods, you’ve used mass (% w/v) concentration (Line 212). Please check it.
Line 353 – “b) cardinal pH values”. Please specify the temperature value.
Line 364 – Please specify the time of incubation. May be 30 min?
Lines 370-372 – May be, it’s better to remove it from the text.
Lines 408-412 – This is a repeat of the Results. I think it’s better to remove it from the text.
Comments on the Quality of English LanguageEnglish language is good. Only minor editing of English language is required.
Author Response
Authors’ Responses to Reviewer 5:
Comments and Suggestions for Authors
The authors investigated the antibacterial potential of three thermophilic bacteria (two Bacillus sp. strains and S. azurea strain) isolated from the Mahallat hot spring in Iran. All three isolates revealed antibacterial activity against control pathogenic strains of S. aureus (Gram-positive) and E. coli (Gram-negative). Their potential antibacterial properties may be useful for possible medical applications in the field of infectious diseases. I do find this work very interesting and valuable. However, there are some issues which should be addressed before publishing.
Specific comments:
- The Title, Lines 2-3 – “Characteristics”, not “Character-istics”.
- Thank you for precise attention. It was corrected.
- Figure 6 – The title of the Y axis: How did you calculate this value? Please add this in the Methods.
- Thanks for your suggestion. It was added to the subsection 2.10 as below and highlighted in yellow in the revised version of the manuscript.
“After measuring the diameter of the halo zones, the maximum values for the bacteria were considered as 100%, with any changes at higher temperatures compared to this value reported as a percentage”.
- Table 2: Columns 3,4 – Mean or maximum values? Please adjust it to the values in Fig. 4.
- Thanks for the comment. In Table 2, the values were related to initial screening and after 48 hours, but in Figure 4, they were checked at different times and there is no discrepancy between the data in Table 2 and Figure 4.
- Line 134 – “TSB”? Please decode it.
- Thank you for the comment. It was corrected and shown in yellow (Page 3).
- Lines 208-209 – “temperatures ranging from 30 to 80° C”. I suppose “from 10 to 80 °C”. See Fig. 5a.
- Thank you for your precise attention. It was properly corrected.
- Line 212 – Is it correct: “%(w/v) NaCl”?
- Thanks for your comment. Yes. It is correct. 20 grams of NaCl were used in 100 mL culture medium, which is 20% by weight/volume.
- Lines 267 – “through 16S rRNA” instead of “through16S rRNA”.
- Thank you for your precise attention. It was properly corrected.
- Lines 301,306 – “lag phase” and “log phase”. Please correct it.
- Thank you for your comment. It was corrected.
- Lines 333-334 – “temperature that Bacillus strains (Kh2 and Kh5) and S. azurea strain Kh3 could tolerate was 75 and 60 °C, respectively”. May be, 60 and 75 °C for Kh2 and Kh5, respectively, and 70 °C for Kh3? See Fig. 5a.
- Thank you for your comment. It was corrected.
- Line 340 – “no considerable growth was detected at pH 9 and 10 for all three strains”. A considerable growth of Kh3 was detected at pH 9 (see Fig. 5b and Line 339). Please correct it.
- Thank you for your comment. It was corrected.
- Lines 345, 347, 348 and 350 – What did you mean: weight (% w/w) or mass (% w/v) concentration of NaCl solution? In Methods, you’ve used mass (% w/v) concentration (Line 212). Please check it.
- Thanks for your comment. It was properly corrected.
- Line 353 – “b) cardinal pH values”. Please specify the temperature value.
- Thanks for your comment. It was properly revised.
- Line 364 – Please specify the time of incubation. May be 30 min?
- Thanks for your comment. It was properly revised.
- Lines 370-372 – May be, it’s better to remove it from the text.
- Thanks for your suggestion. It was removed.
- Lines 408-412 – This is a repeat of the Results. I think it’s better to remove it from the text.
- Thanks for your suggestion. It was removed.
Comments on the Quality of English Language
English language is good. Only minor editing of English language is required.
- Thanks for your feedback. We revised the manuscript again to avoid any English language mistakes.

Reviewer 5 Report
Comments and Suggestions for Authors
This is a well-written paper dealing with the physiological properties and antibacterial potential of hot spring bacteria.
I have only a few comments:
Phylums such as Cyanobacteria, Actinobacteria are not italicised. Italics are used for genera and species. Please correct in the manuscript
Please provide larger images for Figure 1, it is very difficult to recognise
Author Response
Authors’ Responses to Reviewer 6:
Comments and Suggestions for Authors
This is a well-written paper dealing with the physiological properties and antibacterial potential of hot spring bacteria. I have only a few comments:
- Phylums such as Cyanobacteria, Actinobacteria are not italicised. Italics are used for genera and species. Please correct in the manuscript.
- Thanks for your precise attention. They were properly corrected.
- Please provide larger images for Figure 1, it is very difficult to recognize
- Thanks for your suggestion. The size of image was enlarged.

Round 2
Reviewer 1 Report
Comments and Suggestions for Authors
I have gone again through the revised manuscript “Unveiling Antibacterial Potential and Physiological Characteristics of Thermophilic Bacteria Isolated from a Hot Spring in Iran”. I believe that the article still does not meet the standards for publication in the journal Microorganisms. Despite the authors' acknowledgment that 16S rRNA gene amplification and sequence analysis do not offer accurate classification and identification, they did not pursue further genomic analyses due to financial constraints. While I understand that research can be expensive, lack of funds is not a sufficient reason for accepting a work for publication. Furthermore, the presented data remains preliminary with a lack of substantial discussion. Overall, the content lacks significance, scientific soundness, and interest. Although the authors claim to have revised the manuscript to address these issues, very few changes have been made in this revised version. Therefore, I recommend rejecting the article.
Reviewer 2 Report
Comments and Suggestions for Authors
No further comments